# Limit Distributions of Products of Independent and Identically Distributed Random 2 × 2 Stochastic Matrices: A Treatment with the Reciprocal of the Golden Ratio

Santanu Chakraborty

School of Mathematical and Statistical Sciences, University of Texas Rio Grande Valley, 1201 West University Drive, Edinburg, TX 78539-2999, USA; santanu.chakraborty@utrgv.edu

**Abstract:** Consider a sequence $(X_n)_{n \geq 1}$ of i.i.d. $2 \times 2$ stochastic matrices with each $X_n$ distributed as $\mu$. This $\mu$ is described as follows. Let $(C_n, D_n)^T$ denote the first column of $X_n$ and for a given real $r$ with $0 < r < 1$, let $r^{-1}C_n$ and $r^{-1}D_n$ each be Bernoulli distributions with parameters $p_1$ and $p_2$, respectively, and $0 < p_1, p_2 < 1$ . Clearly, the weak limit of the sequence $\mu^n$, namely $\lambda$, is known to exist, whose support is contained in the set of all $2 \times 2$ rank one stochastic matrices. In a previous paper, we considered $0 < r \leq \frac{1}{2}$ and obtained $\lambda$ explicitly. We showed that $\lambda$ is supported countably on many points, each with positive $\lambda$-mass. Of course, the case $0 < r \leq \frac{1}{2}$ is tractable, but the case $r > \frac{1}{2}$ is very challenging. Considering the extreme nontriviality of this case, we stick to a very special such $r$, namely, $r = \frac{\sqrt{5}-1}{2}$ (the reciprocal of the golden ratio), briefly mention the challenges in this nontrivial case, and completely identify $\lambda$ for a very special situation.

**Keywords:** random walk; stochastic matrices; limiting measure; golden ratio

**MSC:** 60B15

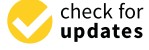



## 1. Introduction

As the title of the paper suggests, the reader can understand that this paper deals with a situation where one considers products of independent and identically distributed random $2 \times 2$ stochastic matrices and their limiting behavior. In other words, here we are considering a probability measure $\mu$ on a collection of $2 \times 2$ stochastic matrices and studying the limiting behavior of the convolution sequence $\mu^n$. To a reader new to this area, the author would like to refer the reader to the book by Hognas and Mukherjea [1]. This book starts from the very basic concepts, such as the definition of a semigroup, topological semigroups, semigroups of matrices, etc., in chapter 1 and then moves forward to more complex concepts, such as probability measures of semigroups, convolution products of probabilities and convergence, random walks on semigroups, random walks on semigroups of nonnegative matrices (and in particular stochastic matrices), etc. The current author collaborated on a few papers in this area [2–6].

For complete understanding of this article, we will go over a few details about convergence of convolution products of probability measures on semigroups of matrices. If $\mathbb{B}$ denotes the collection of Borel subsets of a set $S$, then $P(S)$ can be the set of all regular probability measures $\mu$ on $\mathbb{B}$. Then, denoting the collection of continuous functions on $S$ as $C(S)$, for $\mu, \nu \in P(S)$, and $f \in C(S)$, one defines the following iterated integral:

$$I(f) = \int \int f(xy)\mu(dx)\nu(dy)$$

By the Riesz representation theorem, there exists a unique regular probability measure $\lambda$ such that for any function $f \in C(S)$ with compact support, we have

$$I(f) = \int f d\lambda$$

Then, $\lambda$ is called the convolution of the probability measures $\mu$ and $\nu$. There is a proposition in [1] that shows that for $\mu, \nu \in P(S)$, and $B \in \mathbb{B}$,

$$\mu * \nu(B) = \int \mu(Bx^{-1})\nu(dx) = \int \nu(x^{-1}B)\mu(dx)$$

Having defined the convolution product of regular probability measures on semigroups, one can consider a sequence of regular probability measures $\mu_1, \mu_2, \mu_3, \ldots$, construct a sequence of convolution products of these regular probability measures $\mu_1, \mu_1 \star \mu_2, \mu_1 \star \mu_2 \star \mu_3, \ldots$, and talk about conditions when such convolution sequences will converge. Then, one can specialize to the independent identically distributed situation where for each $i$, we have, $\mu_i = \mu$ for $i = 1, 2, 3, \ldots$. Then, the convolution sequence looks like $\mu^n$ for $n = 1, 2, 3, \ldots$. In all these situations, [1] assumes that $S$ is a locally compact, second countable Hausdorff topological semigroup.

Then, if someone further specializes to the situation when $S$ is a semigroup of nonnegative matrices or say, stochatic matrices of a fixed order $d$, then one considers the usual matrix topology. There have been quite a few papers that study the conditions when the convolution sequence $\mu^n$ converges. Mukherjea [7] first gave conditions when such a sequence converges for i.i.d. $2 \times 2$ stochastic matrices. Then, subsequently such conditions for higher order stochastic matrices were obtained [5,6]. But none of these papers performed detailed study on the nature of the corresponding limiting measures. But motivated by a paper by Chamayou and Letac [8], we have investigated the nature of the limiting measure $\lambda$ for a very special $\mu$ on $2 \times 2$ i.i.d. stochastic matrices.

Before proceeding further, let us denote the probability measure on stochastic matrices of a fixed order $d$ by $\mu$ and its support by $S(\mu)$. So, $S(\mu)$ is a subcollection of stochastic matrices of a fixed order $d$. Thus, for any convolution product $\mu^n$, we will denote its support by $S(\mu^n)$ and the support of the limiting measure $\lambda$ (if it exists) by $S(\lambda)$.

If we denote the closure of an arbirary set $E$ by $\overline{E}$, then

$$S(\mu^n) = \overline{\{A_1 A_2 \cdots A_n \mid \text{for each } i, \ A_i \in S(\mu), \ 1 \leq i \leq n\}}$$

where $n$ is a positive integer and

$$\mathcal{S} = \overline{\cup_{n=1}^{\infty} S(\mu^n)}$$

Also, denote $\mathbb{P}$ to be the set of $d \times d$ strictly positive stochastic matrices in $\mathcal{S}$.

Chamayou and Letac [8] proved that if $(X_n)_{n \geq 1}$ is a sequence of $d \times d$ i.i.d. stochastic matrices such that $P(\min_{i,j}(X_1)_{ij} = 0) < 1$, then $Y = \lim_{n \to \infty} X_n X_{n-1} \cdots X_1$ exists almost surely and $P(Y \text{ has rank } 1) = 1$; furthermore, if for any Borel $B$ of $d \times d$ stochastic matrices (with usual $R^{d^2}$-topology), we denote $\mu(B) = P(X_1 \in B)$ and $\lambda(B) = P(Y \in B)$, and then $\lambda$ is the unique solution of the convolution equation $\lambda \star \mu = \lambda$.

Then, in [2], we noted that this wonderful result of Chamayou and Letac also holds under the (slightly weaker) condition that $\mu^m(\mathbb{P}) > 0$ for some positive integer $m$ (as opposed to just 1, instead of $m$, taken in [8]) where $\mu^m$ is the distribution of the product $X_m \cdots X_1$ and $\mathbb{P}$ is the set of $d \times d$ strictly positive stochastic matrices. The reason is as follows: the Chamayou and Letac result shows that under the weaker condition, the subsequence $Y_{nm} = X_{nm} X_{nm-1} \cdots X_1$ converges almost surely to some $d \times d$ rank one stochastic matrix, $Y_0$, and consequently, any subsequence $X_{n_k} X_{n_k-1} \cdots X_1$ with $n_k > s_k m$ (for some $s_k$) will also converge almost surely to a $d \times d$ stochastc matrix $VY_0(= Y_0$, as $Y_0$ has rank one), where $V$ is a limit point of the product subsequence $X_{n_k} X_{n_k-1} \cdots X_{s_k m+1}$. This establishes our observation.

Next we mention below some situations when $S(\lambda)$ consists of all rank one matrices:

Situation 1: If $(X_i)_{i\geq 1}$, as before, is i.i.d. $d \times d$ stochastic matrices such that for some positive integer $m \geq 1$,

$$\mu^m(\mathbb{P}) > 0 \qquad (1)$$

then the sequence $\mu^n$, where $\mu(B) = P(X_1 \in B)$ for Borel sets $B$ of $d \times d$ stochastic matrices, converges weakly to a probability measure $\lambda$ and $S(\lambda)$ consists of all rank one stochastic matrices in $\mathcal{S}$ such that $\lambda(\mathbb{P}) > 0$.

Situation 2: When $\lambda$ is the weak limit of $(\mu^n)_{n \geq 1}$ and $\mathcal{S}$ contains a rank one matrix, then the support of $\lambda$, $S(\lambda)$ consists of all rank one stochastic matrices in $\mathcal{S}$. This is an algebraic fact for the support of an idempotent probability measure (note that $\lambda = \lambda \star \lambda$; see [1]).

In the same paper, Chamayou and Letac (see also [9]) tried to identify $\lambda$ in the case when the rows of $X_1$ above are independent, and for $1 \leq i \leq d$, the $i$-th row of $X_1$ has Dirichlet distribution with positive parameters $\alpha_{i1}, \alpha_{i2}, \ldots, \alpha_{id}$, and they were successful in the case when $\sum_{j=1}^{d} \alpha_{ij} = \sum_{j=1}^{d} \alpha_{ji}$, $1 \leq i \leq d$. Indeed, there are only very few (other than those given in [8–10]) examples in the literature even for $2 \times 2$ stochastic matrices when the limit distribution $\lambda$ has been identified completely in the above context. Our paper [2] is an example.

In [2], we considered $2 \times 2$ i.i.d. stochastic matrices $(X_n)_{n \geq 1}$ with $X_n = \begin{pmatrix} C_n & 1 - C_n \\ D_n & 1 - D_n \end{pmatrix}$, each $X_n$ is distributed as $\mu$ and $r^{-1}C_n$ and $r^{-1}D_n$ are each Bernoulli distributions (with possibly different parameters $p_1$ and $p_2$, $0 < p_1, p_2 < 1$) for a real $r$ satisfying $0 < r \leq 1$. Our goal was to identify $\lambda$, the distribution of $\lim_{n \to \infty} X_n X_{n-1} \cdots X_1$. Clearly, there are exactly four matrices in the support of $\mu$, each with positive mass. It is well known that that $\mu^n$ converges weakly to a limiting measure $\lambda$ and the support of $\lambda$ consists of rank one matrices. In particular, if $r$ equals 1, the support of $\lambda$ has exactly two matrices, namely, $\begin{pmatrix} 0 & 1 \\ 0 & 1 \end{pmatrix}$ and $\begin{pmatrix} 1 & 0 \\ 1 & 0 \end{pmatrix}$. In [2], a complete solution is given to the problem for $0 < r \leq \frac{1}{2}$ and also for $r = 1$.

The situation $\frac{1}{2} < r < 1$ is much more challenging. Before explaining where the challenge lies, let us make the following convention:

From now on, we will often denote the matrix $\begin{pmatrix} x & 1 - x \\ x & 1 - x \end{pmatrix}$ by simply $x$ when there is no fear of confusion. Thus, for the limiting measure $\lambda$, $\lambda(x)$ will mean $\lambda\begin{pmatrix} x & 1 - x \\ x & 1 - x \end{pmatrix}$ and if we write that the support of $\lambda$, $S(\lambda)$ is contained in $[0, 1]$, then this means the following:

$$S(\lambda) \subset \left\{ \begin{pmatrix} x & 1 - x \\ x & 1 - x \end{pmatrix} : 0 \leq x \leq 1 \right\}$$

Now, we are going to explain why the case $\frac{1}{2} < r < 1$ is more challenging. Although we find it quite easy to observe that $\lambda(0)$ and $\lambda(r)$ have the same expressions as in the previous case, it is indeed hard to exhibit a point in $(0, r)$ with positive $\lambda$-mass.

However, there is a special situation when things are more tractable, namely, $r = \frac{\sqrt{5}-1}{2}$ (the reciprocal of the golden ratio). We denote this special $r$ as $r_g$. Notice that $r_g$ satisfies the equation $r_g^2 + r_g - 1 = 0$. Using this equation extensively, we completely solve for $\lambda$ in this particular situation. It can be seen that although this is just one case, the proof is highly nontrivial. According to the author, the reason why $r_g$ works for us is because of the fact that $\lambda(1 - r_g)$ could be found out easily and so this technique of proof worked.

It may be mentioned here that there have been numerous studies in the literature involving the golden ratio. One very recent study invloving golden ratio is in the context of machine learning [11].

As in the case of $0 < r \leq \frac{1}{2}$, here also $\lambda$ is discrete with masses at countably many points. Our main theorem appears in Section 4.

One gets a feeling that for any other $r$ satisfying $\frac{1}{2} < r < 1$, finding the value of $\lambda(1-r)$ itself will be a challenge, making it quite nontrivial. Thus, for a general $\frac{1}{2} < r < 1$, a different technique of proof might be needed to obtain a complete solution.

In the next section (Section 2), we describe our set up, state the results proved in [2] for $0 < r \leq \frac{1}{2}$, and briefly discuss the more challenging situation $\frac{1}{2} < r < 1$. In Section 3, we focus on $r = r_g = \frac{\sqrt{5}-1}{2}$ (reciprocal of the golden ratio) and prove two important propostions. We prove our main theorem and a series of lemmas leading to it in Section 4. We have some concluding remarks and comments in Section 5.

## 2. Preliminaries

In our case, we are considering the case of a probability measure $\mu$ on $2 \times 2$ stochastic matrices. $S(\mu)$ denotes its support, which is a subcollection of $2 \times 2$ stochastic matrices. $S(\mu^n)$ denotes the support of $\mu^n$ where $\mu^n$ is the convolution sequence. As pointed out in [7], $\mu^n$ converges if and only if $S(\mu)$ is not a singleton:

$$S(\mu) \neq \left\{ \begin{pmatrix} 0 & 1 \\ 1 & 0 \end{pmatrix} \right\}$$

And in case there is a strictly positive matrix in $S(\mu)$, then the support $S(\lambda)$ of the limiting measure $\lambda$ consists of rank one matrices. Our special case satisfies that condition:

We consider $2 \times 2$ i.i.d. stochastic matrices $(X_n)_{n \geq 1}$ with $X_n = \begin{pmatrix} C_n & 1 - C_n \\ D_n & 1 - D_n \end{pmatrix}$, such that each $X_n$ is distributed as $\mu$. Also, assume that for a given $r$ with $0 < r \leq 1$, both $r^{-1}C_n$ and $r^{-1}D_n$ are Bernoulli distributions with parameters $p_1$ and $p_2$ respectively .

Then, the support of $\mu$ has exactly four matrices as $S(\mu)$ is given by:

$$S(\mu) = \left\{ \begin{pmatrix} 0 & 1 \\ 0 & 1 \end{pmatrix}, \begin{pmatrix} 0 & 1 \\ r & 1-r \end{pmatrix}, \begin{pmatrix} r & 1-r \\ 0 & 1 \end{pmatrix}, \begin{pmatrix} r & 1-r \\ r & 1-r \end{pmatrix} \right\}$$

Let the $\mu$-masses at these points be denoted by $p_{00}$, $p_{01}$, $p_{10}$, $p_{11}$ respectively so that $p_{00} + p_{01} = 1 - p_1$, $p_{00} + p_{10} = 1 - p_2$, $p_{10} + p_{11} = p_1$ and $p_{01} + p_{11} = p_2$.

Let $\lambda$ be the distribution of $\lim_{n \to \infty} X_n X_{n-1} \cdots X_1$.

In case $r$ equals 1, one can easily observe that $\lambda$ is a Bernoulli distribution with parameters entirely dependent on the probability mass function of $\mu$, namely,

$$\lambda(0) = \frac{p_{00} + p_{01}}{1 - p_{10} + p_{01}}$$

This follows by solving for $\lambda(0)$ and $\lambda(1)$ in the convolution equation $\lambda \star \mu = \lambda$.

For $0 < r < 1$, the support of $\mu^n$, $S(\mu^n)$ and consequently $\mathcal{S}$ is contained in the set

$$\left\{ \begin{pmatrix} x & 1-x \\ y & 1-y \end{pmatrix} : 0 \leq x \leq r, 0 \leq y \leq r \right\}$$

This can be proved using induction on $n$. One assumes up to some positive integer $l$ and proves for $l + 1$ by noticing that when one multiplies a matrix in $S(\mu^l)$ by a matrix in $S(\mu)$, the entries in the product matrix satisfies the condition that each entry in the first column is between 0 and $r$ because each entry in the first column of the matrices from $S(\mu^l)$ and $S(\mu)$ is so.

Also, since the relation $\lambda \star \mu = \lambda$ holds, the support of $\lambda$, namely, $S(\lambda)$ consists of all rank one matrices in $\mathcal{S}$. As a result, $S(\lambda) \subset \{x : 0 \leq x \leq r\}$, where $x$ stands for $\begin{pmatrix} x & 1-x \\ x & 1-x \end{pmatrix}$. Moreover, exploiting the identity $\lambda \star \mu = \lambda$, we have

$$\lambda(0) = \frac{p_{00}}{1 - p_{10}}, \quad \lambda(r) = p_{11} + \lambda(0)p_{01} = \frac{p_{11}(1 - p_{10}) + p_{00}p_{01}}{1 - p_{10}}$$

and for other points $x$ with $0 < x < r$ with positive $\lambda$-masses, we have

$$\lambda(x) = \lambda(r^{-1}x)p_{10} + \lambda(1 - r^{-1}x)p_{01} \tag{2}$$

Next, we state the results proved in [2] for $0 < r \leq \frac{1}{2}$:

*2.1. Case: $0 < r \leq \frac{1}{2}$*

First of all, we introduce some notations. For each $i \geq 1$, define

$$A_i = \left\{ \sum_{j=1}^{k}(-1)^{j-1}r^{i_j} : 1 \leq i_1 < i_2 < i_3 < \cdots < i_k = i, k \leq i \right\}, \quad A = \cup_{i=1}^{\infty}A_i$$

We have two propositions for taking care of the cases $0 < r < \frac{1}{2}$ and $r = \frac{1}{2}$:

**Proposition 1.** *For $0 < r < \frac{1}{2}$, we have the following:*

(i)  *For every positive integer $i \geq 1$, $|A_i| = 2^{i-1}$ and each point in $A_i$ has positive $\lambda$-mass. These are the only points of degree $i$ in the support of $\lambda$ with positive $\lambda$-mass.*

(ii)  *Each such point has $\lambda$-measure equal to $\lambda(r)p_{10}^{i-1-k}p_{01}^{k}$. For every $i > 1$, $\lambda(A_i) = \lambda(r)[p_{10} + p_{01}]^{i-1}$.*

(i)  $\lambda(0) + \sum_{i=1}^{\infty}\lambda(A_i) = \lambda(0) + \lambda(r) \cdot \left[ \sum_{i=1}^{\infty}(p_{10} + p_{01})^{i-1} \right] = 1$.

**Proposition 2.** *For $r = \frac{1}{2}$, we have the following:*

(i)  *The only points that have positive $\lambda$-masses are the dyadic rationals in $[0, \frac{1}{2}]$. Thus, for every $i$, there are exactly $2^{i-2}$ dyadic rationals of the form $\frac{k}{2^i}$ with $k \leq 2^{i-1}$ and $k$ odd with positive $\lambda$-mass. $A_i$ consists of exactly these points. Also, $|A_i| = 2^{i-2}$.*

(ii)  *A typical point in $A_i$ has $\lambda$-measure equal to $\lambda\left(\frac{1}{2}\right)(p_{10} + p_{01})p_{10}^{i-1-k}p_{01}^{k-1}$ for some positive integer $k$. For every $i > 1$, $\lambda(A_i) = \lambda\left(\frac{1}{2}\right)[p_{10} + p_{01}]^{i-1}$.*

(iii)  *The sum of the $\lambda$-masses of all dyadic rationals in $\left[0, \frac{1}{2}\right]$ along with the $\lambda$-mass at zero equals 1. Equivalently, $\lambda(0) + \sum_{i=1}^{\infty}\lambda(A_i) = \lambda(0) + \lambda\left(\frac{1}{2}\right) \cdot \left[ \sum_{i=1}^{\infty}(p_{10} + p_{01})^{i-1} \right] = 1$*

The case $\frac{1}{2} < r < 1$ turns out to be quite nontrivial. We briefly introduce that case below:

*2.2. Case: $\frac{1}{2} < r < 1$*

The case $\frac{1}{2} < r < 1$ is distinctly different from the case $r < \frac{1}{2}$ because now we have $1 - r < r$. Since for each $r$, $\lambda$ has masses at 0 and $r$, it is not absolutely continuous for any $r$. Now, suppose we continue with the same notation of $A$ introduced in the case $0 < r \leq \frac{1}{2}$. Thus, $A = \cup_{i=1}^{\infty}A_i$ where, for every positive integer $i$,

$$A_i = \left\{ \sum_{j=1}^{k}(-1)^{j-1}r^{i_j} : 1 \leq i_1 < i_2 < i_3 < \cdots < i_k = i, k \leq i \right\}$$

It then easily follows that each of these points in $A$ also has positive mass even in the case $\frac{1}{2} < r < 1$. However, it is indeed a challenge to calculate $\lambda$-masses at these points.

Also, since $1 - r \in (0, r)$, it is natural to have points of the form $1 + \sum_{j=1}^{k}(-1)^{j}r^{i_j}$, $1 \leq i_1 < i_2 < i_3 < \cdots < i_k = i, k \leq i$ for any positive integer $i$ in the interval $(0, r)$ (to see this, notice that $r^{i_1} > \sum_{j=2}^{k}(-1)^{j}r^{i_j}$). Accordingly, define $A^* = \cup_{i=1}^{\infty}A_i^*$, where

$$A_i^* = \left\{ 1 + \sum_{j=1}^{k}(-1)^{j}r^{i_j} : 1 \leq i_1 < i_2 < i_3 < \cdots < i_k = i, k \leq i \right\}$$

Recall that, for $0 < r \leq \frac{1}{2}$, each point in $A$ has positive $\lambda$-mass and each point in $A^*$ is outside $(0, r)$ and has zero $\lambda$-mass.

For $\frac{1}{2} < r < 1$, of course, each polynomial in $A$ is in $(0, r)$. But, although some polynomials in $A^*$ are numerically less than $r$, it is not easy to see which of these points have positive $\lambda$-masses. Clearly, some polynomials in $A_i^*$ are outside $(0, r)$ and have zero $\lambda$-measure if $i$ is large enough. For example, for a fixed $r$, it is possible to get a positive integer $m > 1$ such that $1 - r^m \geq r > 1 - r^{m-1}$. Next, consider $i_1 = l$ with $l \geq m$ for a polynomial $1 + \sum_1^k (-1)^j r^{i_j}$ in $A_i^*$ with $1 \leq i_1 < i_2 < i_3 < \cdots < i_k = i$, $k \leq i$. Then, this polynomial is greater than or equal to $1 - r^m + \sum_2^k (-1)^j r^{i_j}$, which is obviously greater than $r$ and has $\lambda$-measure zero. But, it is a possibility that some points in $A^*$ could have positive $\lambda$-masses too.

Recall the very special $r$, $r = r_g$, the reciprocal of the golden ratio. We know $r_g$ satisfies the equation $r_g^2 + r_g - 1 = 0$ and $1 - r_g$ actually equals $r_g^2$, whose $\lambda$-measure can be found out easily. The next two sections deal with this special case.

## 3. $r = r_g$: Main Results

In this section and also in the next section, we deal with $r = r_g = \frac{\sqrt{5}-1}{2}$ unless stated otherwise. This is a very special case of $\frac{1}{2} < r < 1$. Note that $r_g$ is the reciprocal of the golden ratio and is the positive solution of the equation $r^2 + r - 1 = 0$. To avoid dealing with too many radical signs and complicating matters, we will continue to use $r_g$ in these two sections for this particluar choice of $r$.

**Remark 1.** *A polynomial $1 + \sum_1^k (-1)^j r_g^{i_j}$ in $A_i^*$ with $1 \leq i_1 < i_2 < i_3 < \cdots < i_k = i$, $k \leq i$ and $i_1 \geq 2$ has zero $\lambda$-measure.*

This is because, $1 - r_g^2 = r_g$ implies that such a polynomial is greater than $r_g$ in magnitude. However, for $i_1 = 1$, such a polynomial may have positive $\lambda$-measure as well.

In order to notice this, first observe that, $\lambda(1 - r_g) > 0$. This is because, using (2), we have

$$\lambda(1 - r_g) = \lambda(r_g^2) = \lambda(r_g) p_{10} + \lambda(1 - r_g) p_{01}$$

implying that $\lambda(1 - r_g) = \frac{p_{10}}{1 - p_{01}} \lambda(r_g)$ where $\lambda(r_g)$ is already known.

Next, consider a nontrivial example, say, the polynomial $1 - r_g + r_g^2 - r_g^3$. Using (2) repeatedly and Remark 2, we find that its $\lambda$- measure equals

$$\lambda(r_g^2) p_{10}^2 p_{01} + \lambda(1 - r_g^2) p_{10} p_{01}^2 = \lambda(1 - r_g) p_{10}^2 p_{01} + \lambda(r_g) p_{10} p_{01}^2$$

implying that the polynomial under consideration has non-zero $\lambda$-measure. Since we know $\lambda(r_g)$ and $\lambda(1 - r_g)$, it is possible to find out $\lambda(1 - r_g + r_g^2 - r_g^3)$ explicitly.

But, this is only a particular example. Can we make a general observation? Yes. Look at the following result.

**Proposition 3.** *Any polynomial in $A^*$ either has $\lambda$-measure 0 or can be written as a polynomial in $A$.*

**Proof.** To fix ideas, we assume that our polynomial in $A^*$ is $1 + \sum_{j=1}^k (-1)^j r_g^{i_j}$ with $1 \leq i_1 < i_2 < i_3 < \cdots < i_k = i$ and $k \leq i$. Because of Remark 3.0, we can assume that $i_1 = 1$. Then, we consider the following cases:

Case 1: $i_j = j$ for $j = 2, 3, \ldots, k$.

Then, the given polynomial equals $1 - r_g + r_g^2 - \cdots + (-1)^k r_g^k$

Subcase 1: $k$ is even, say, $k = 2m$. Then, the above polynomial equals $1 - r_g + r_g^2 - \cdots + r_g^{2m}$. Notice that $r_g^j - r_g^{j+1} = r_g^{j+1} - r_g^{j+3}$ for $j = 0, 1, 2, \ldots$. Thus, the given polynomial

equals $r_g - r_g^3 + r_g^3 - r_g^5 + \cdots + r_g^{2m-1} - r_g^{2m+1} + r_g^{2m}$ which equals $r_g - r_g^{2m+1} + r_g^{2m} > r_g$. So, it has $\lambda$-measure 0.

<u>Subcase 2:</u> $k$ is odd, $k = 2m + 1$. Then, the above polynomial equals $1 - r_g + r_g^2 - \cdots + r_g^{2m} - r_g^{2m+1}$. Once again recall that $r_g^j - r_g^{j+1} = r_g^{j+1} - r_g^{j+3}$ for $j = 0, 1, 2, \ldots$. So, the given polynoimal equals $r_g - r_g^3 + r_g^3 - r_g^5 + \cdots + r_g^{2m-1} - r_g^{2m+1} + r_g^{2m+1} - r_g^{2m+3} = r_g - r_g^{2m+3}$. And it is a polynomial in $A$.

<u>Case 2:</u> There exists an $l$ such that $i_l > l$ and $i_j = j$ for $j < l$. Then, the given polynomial equals $1 - r_g + r_g^2 - \cdots + (-1)^{l-1} r_g^{l-1} + \sum_{j=l}^{k} (-1)^j r_g^{i_j}$.

<u>Subcase 1:</u> $l$ is even, say, $l = 2m$. Then, the polynomial equals $1 - r_g + r_g^2 - \cdots - r_g^{2m-1} + \sum_{j=2m}^{k} (-1)^j r_g^{i_j}$. Again, we use $r_g^j - r_g^{j+1} = r_g^{j+1} - r_g^{j+3}$ for $j = 0, 1, 2, \ldots$ so that the given polynomial equals $r_g - r_g^3 + r_g^3 - r_g^5 + \cdots + r_g^{2m-1} - r_g^{2m+1} + \sum_{j=2m}^{k} (-1)^j r_g^{i_j}$. If $i_{2m} = 2m + 1$, then this polynomial equals $r_g - r_g^3 + r_g^3 - r_g^5 + \cdots + r_g^{2m-1} - r_g^{2m+1} + r_g^{2m+1} + \sum_{j=2m+1}^{k} (-1)^j r_g^{i_j}$, which equals $r_g + \sum_{j=2m+1}^{k} (-1)^j r_g^{i_j}$. This is, of course, a polynomial in $A$. On the other hand, if $i_{2m} > 2m + 1$, then the above polynomial equals $r_g - r_g^{2m+1} + \sum_{j=2m}^{k} (-1)^j r_g^{i_j}$. Once again, it is a polynomial in $A$.

<u>Subcase 2:</u> $l$ is odd, say, $l = 2m + 1$. Then, the given polynomial equals $1 - r_g + r_g^2 - \cdots - r_g^{2m-1} + r_g^{2m} + \sum_{j=2m+1}^{k} (-1)^j r_g^{i_j}$. Applying once again $r_g^j - r_g^{j+1} = r_g^{j+1} - r_g^{j+3}$ for $j = 0, 1, 2, \ldots$, this polynomial equals $r_g - r_g^3 + r_g^3 - r_g^5 + \cdots + r_g^{2m-1} - r_g^{2m+1} + r_g^{2m} + \sum_{j=2m+1}^{k} (-1)^j r_g^{i_j}$. This simplifies to $r_g - r_g^{2m+1} + r_g^{2m} + \sum_{j=2m+1}^{k} (-1)^j r_g^{i_j} = r_g + r_g^{2m+2} + \sum_{j=2m+1}^{k} (-1)^j r_g^{i_j}$. If $i_{2m+1} = 2m + 2$, then the above equals $r_g + \sum_{j=2m+2}^{k} (-1)^j r_g^{i_j} > r_g$. So, it has $\lambda$-measure zero. If $i_{2m+1} > 2m + 2$, then the given polynomial equals $r_g + r_g^{2m+2} + \sum_{j=2m+1}^{k} (-1)^j r_g^{i_j}$ which is same as $r_g + r_g^{2m+2} - r_g^{i_{2m+1}} + \sum_{j=2m+2}^{k} (-1)^j r_g^{i_j} > r_g$. So, it has $\lambda$-measure equal to zero. $\square$

**Remark 2.** *Because of the above proposition, it is good enough to consider only polynomials in $A$. We will rather consider the same polynomials as in the case $0 < r < \frac{1}{2}$ and will try to work out their $\lambda$-measures.*

*We have seen in Section 2 that the number of elements in $A_n$ equals $2^{n-1}$. But, because of the relationship $1 - r_g = r_g^2$ in the current situation, there will be redundancy and all polynomials are not distinct. So, we will see that we need to consider at most $2^{n-2}$ elements from $A_n$ for each $n \geq 3$:*

**Proposition 4.** *There are at most $2^{n-2}$ distinct elements in $A_n$ for each $n \geq 3$.*

**Proof.** Once again, the identity $1 - r_g = r_g^2$ has a big role to play. For $n = 1, 2, 3$ or $4$, it is trivial to observe. For general $n$, first notice that $r_g^n = r_g^{n-2} - r_g^{n-1}$, and so $r_g^n$ can be considered to be in $A_{n-1}$. More generally, define

$$Q_n = \left\{ r_g^n \right\} \cup \left\{ \sum_{j=1}^{k} (-1)^{j-1} r_g^{i_j} : 1 \leq i_1 < \cdots < i_{k-1} < i_k; \ i_{k-1} < n, \ i_k = n; \ k < n; \ n - i_{k-1} \geq 2 \right\}$$

and

$$R_n = \left\{ \sum_{j=1}^{k} (-1)^{j-1} r_g^{i_j} : 1 \leq i_1 < \cdots < i_{k-1} < i_k; \ i_{k-1} = n - 1, \ i_k = n, \ k \leq n \right\}$$

Let $Q = \cup_{n=3}^{\infty} Q_n$ and $R = \cup_{n=3}^{\infty} R_n$. Then, observe that each polynomial in $Q$ is numerically equal to a polynomial in $R$ of less degree.

We see this as follows:

Consider an $n > 2$. Take a polynomial in $Q_n$. If it is $r_g^n$, we have already provided the argument, that is, $r_g^n = r_g^{n-2} - r_g^{n-1} \in R_{n-1}$. Otherwise, consider a typical element from $Q_n$, say, $r_g^{i_1} - r_g^{i_2} + \cdots + (-1)^{k-1} r_g^{i_{k-1}} + (-1)^k r_g^n$ with $1 \le i_1 < i_2 < \cdots < n$ and for some $k < n$. If $n - i_{k-1} = 2$, then $r_g^{i_{k-1}} - r_g^n = r_g^{n-2} - r_g^n = r_g^{n-1} = r_g^{n-3} - r_g^{n-2}$. As a result, the given polynomial equals $r_g^{i_1} - r_g^{i_2} + \cdots + + (-1)^{k-1} r_g^{n-3} + (-1)^k r_g^{n-2}$. So, it is a polynomial in $R$ of less degree $(n-2)$. On the other hand, if $n - i_{k-1} > 2$, then $r_g^{i_{k-1}} - r_g^n = r_g^{i_{k-1}} - r_g^{n-2} + r_g^{n-1}$. So, the given polynomial equals $r_g^{i_1} - r_g^{i_2} + \cdots + (-1)^{k-1} r_g^{i_{k-1}} + (-1)^k r_g^{n-2} + (-1)^{k+1} r_g^{n-1}$. Once again, this is a polynomial in $R$ of less degree $(n-1)$.

It is clear that for each $n$, $A_n = Q_n \cup R_n$ and hence $A = Q \cup R$. So, because of this observation, the only polynomials in $A$ that can be considered for $\lambda$-mass calculation are the ones in $R$. Also, it follows that for $n \ge 3$, $R_n$ has at most $2^{n-2}$ distinct polynomials. Consequently, $A_n$ also has at the most $2^{n-2}$ distinct elements and the proposition follows.  $\square$

**Remark 3.** *Thus, for each n, we have fewer polynomials of degree n compared to the situation* $0 < r < \frac{1}{2}$.

Now, it is time we prove our main theorem. We prove it in the next section.

## 4. $r = r_g$: Proof of the Main Theorem

Here is our main theorem:

**Theorem 1.** *Consider* $r = r_g = \frac{\sqrt{5}-1}{2}$. *Then*

$$\lambda(0) + \lambda(r_g) + \lambda(r_g^2) + \lambda(r_g - r_g^2) + \lambda(R) = 1$$

*where* $R = \cup_{n=3}^{\infty} R_n$ *with*

$$R_n = \left\{ \sum_{j=1}^{k} (-1)^{j-1} r_g^{i_j} : 1 \le i_1 < i_2 < \cdots < i_{k-2} < i_{k-1} < i_k; i_{k-1} = n-1, i_k = n, k \le n \right\}$$

First, notice that, using (2), it follows that $\lambda(r_g^2) = \frac{p_{10}}{1-p_{01}} \lambda(r_g)$ and $\lambda(r_g - r_g^2) = \left( \frac{p_{10}^2}{1-p_{01}} + p_{01} \right) \lambda(r_g)$. Thus, in order to prove the theorem, it is enough to prove:

$$\lambda(R) = \frac{p_{10} + p_{01}}{1 - p_{10} - p_{01}} \lambda(r_g - r_g^2) \tag{3}$$

because then,

$$\lambda(r_g - r_g^2) + \lambda(R) = \left( 1 + \frac{p_{10} + p_{01}}{1 - p_{10} - p_{01}} \right) \lambda(r_g - r_g^2)$$

As a result,

$$\lambda(0) + \lambda(r_g) + \lambda(r_g^2) + \lambda(r_g - r_g^2) + \lambda(R) = \lambda(0) + \frac{1}{1 - p_{10} - p_{01}} \lambda(r_g)$$

But, recall from Section 3:

$$\lambda(0) = \frac{p_{00}}{1 - p_{10}}, \quad \lambda(r_g) = p_{11} + \lambda(0) p_{01} = \frac{p_{11}(1 - p_{10}) + p_{00} p_{01}}{1 - p_{10}}$$

This implies that

$$\lambda(0) + \frac{1}{1 - p_{10} - p_{01}} \lambda(r_g) = 1$$

This is the reason that it is good enough to prove (3). For this, we proceed as follows.

First of all, notice that $R_3 = \{r_g^2 - r_g^3, r_g - r_g^2 + r_g^3\}$, $R_4 = \{r_g^3 - r_g^4, r_g^2 - r_g^3 + r_g^4, r_g - r_g^3 + r_g^4, r_g - r_g^2 + r_g^3 - r_g^4\}$ etc. and in general

$$R_n = \{r_g^{n-1} - r_g^n, r_g^{n-2} - r_g^{n-1} + r_g^n, \ldots, r_g - r_g^{n-1} + r_g^n, \ldots, r_g - r_g^2 + \cdots + r_g^{n-1} - r_g^n\}$$

Next, we introduce some notations for any $0 < r < 1$.

Define $g : R \to R$ and $f_j : R \to R$ for every positive integer $j$ as follows: $g(p) = rp$ and $f_j(p) = r^j - p$. Thus, $R_3 = \{f_2(r^3), f_1 f_2(r^3)\}$, $R_4 = \{f_3(r^4), f_2 f_3(r^4), f_1 f_3(r^4), f_1 f_2 f_3(r^4)\}$ etc., and in general,

$$R_n = \{f_{n-1}(r^n), f_{n-2} f_{n-1}(r^n), \ldots, f_1 f_{n-1}(r^n), \ldots, f_1 f_2 \cdots f_{n-1}(r^n)\}$$

We further define operators $F_j$ for $j \geq 2$ on $R$ as follows: $F_2 = \{f_2, f_1 f_2\}$, $F_3 = \{f_3, f_2 f_3, f_1 f_3, f_1 f_2 f_3\}$ etc., and in general,

$$F_{n-1} = \{f_{n-1}, f_{n-2} f_{n-1}, \ldots, f_1 f_{n-1}, \ldots, f_1 f_2 \cdots f_{n-1}\}$$

Thus, $R_j = F_{j-1}(r^j)$ for $j = 3, 4, \ldots$ and

$$F_{j-1}(p) = \{f_{j-1}(p), f_{j-2} f_{j-1}(p), \ldots, f_1 f_{j-1}(p), \ldots, f_1 f_2 \cdots f_{j-1}(p)\}$$

In general, one would anticipate $|F_2(p)| = 2$, $|F_3(p)| = 4$, $\ldots$, $|F_{j-1}(p)| = 2^{j-2}$. But, for $r = r_g$, equality is replaced by $\leq$ for some $p$s.

Now, in order to prove (3), we will use a series of Lemmas 1–5. Lemma 1 identifies that connsecutive $R_i$s have nonempty overlaps for $i \geq 3$, Lemma 2 evaluates the cardinality of the consecutive overlaps, Lemma 3 evaluates the cardinality of the consecutive differences, Lemma 4 calculates the $\lambda$- measures of these differences, and, finally, Lemma 5 puts them together to evaluate the $\lambda$-measure of $R$ thereby proving (3). Thus, once Lemmas 1–5 are proved, (3) is proved and the proof of the theorem is complete.

**Lemma 1.** *Consecutive $R_i$s ($R_i$ and $R_{i+1}$) have nonempty intersections for $i \geq 3$. In fact, $R_4 \cap R_3 = \varnothing$ but $R_{j+1} \cap R_j \neq \varnothing$ for $j > 3$*

**Proof.** It is trivial to observe that $R_4 \cap R_3 = \phi$. Now, notice that $r_g^2 - r_g^4 + r_g^5$, $r_g - r_g^2 + r_g^4 - r_g^5 \in R_5 \cap R_4$ because $r_g^2 - r_g^4 + r_g^5 = r_g - r_g^2 + r_g^5 = r_g - r_g^2 + r_g^3 - r_g^4 \in R_4$ and automatically, $r_g - r_g^2 + r_g^4 - r_g^5 = r_g - (r_g - r_g^2 + r_g^3 - r_g^4) = r_g^2 - r_g^3 + r_g^4 \in R_4$. Thus, $R_5 \cap R_4 = F_2(r_g^4 - r_g^5)$ and $|R_5 \cap R_4| = 2$. In general, $R_{j+1} \cap R_j \supseteq F_{j-2}(r_g^j - r_g^{j+1}) \cup F_{j-4}(r_g^j - r_g^{j+1})$ for $j \geq 6$. In fact, we can show that for positive integers $k \geq 3$

$$R_{2k-1} \cap R_{2k-2} = F_{2k-4}(r_g^{2k-2} - r_g^{2k-1}) \cup F_{2k-6}(r_g^{2k-2} - r_g^{2k-1}) \cup \cdots \cup F_2(r_g^{2k-2} - r_g^{2k-1})$$

$$R_{2k} \cap R_{2k-1} = F_{2k-3}(r_g^{2k-1} - r_g^{2k}) \cup F_{2k-5}(r_g^{2k-1} - r_g^{2k}) \cup \cdots \cup F_3(r_g^{2k-1} - r_g^{2k})$$

So, Lemma 1 is proved. $\square$

**Lemma 2.** *For $i \geq 4$, $|R_i \cap R_{i+1}|$s are evaluated upper bounds for $|R_{i+1} - R_i|$ are determined as follows:*

*For $k \geq 3$, we have,*

$$|R_{2k-1} \cap R_{2k-2}| = \frac{2}{3}(2^{2k-4} - 1), \quad |R_{2k} \cap R_{2k-1}| = \frac{4}{3}(2^{2k-4} - 1)$$

*so that*

$$|R_{2k-1} - R_{2k-2}| \le \frac{2^{2k-2}+2}{3}, \quad |R_{2k} - R_{2k-1}| \le \frac{2^{2k-1}+4}{3}$$

**Proof.** From Lemma 1, it follows that $|R_5 \cap R_4| = 2$ implying $|R_5 - R_4| \le 2^3 - 2 = 6$, $|R_6 \cap R_5| = 4$ implying $|R_6 - R_5| \le 2^4 - 4 = 12$.

In general, notice that for $k \ge 4$,

$$|F_{2k-2l}(r_g^{2k-2l+2} - r_g^{2k-2l+3})| = 2^{2k-2l-1}$$

and

$$|F_{2k-2l+1}(r_g^{2k-2l+3} - r_g^{2k-2l+4})| = 2^{2k-2l}$$

for $2 \le l \le k - 1$. Also,

$$|R_{2k-1} \cap R_{2k-2}| = 2^{2k-5} + 2^{2k-7} + \cdots + 2 = \frac{2}{3}\left(2^{2k-4} - 1\right)$$

implying that $|R_{2k-1} - R_{2k-2}| \le 2^{2k-3} - \frac{2}{3}\left(2^{2k-4} - 1\right) = \frac{2^{2k-2}+2}{3}$ and

$$|R_{2k} \cap R_{2k-1}| = 2^{2k-4} + 2^{2k-6} + \cdots + 2^2 = \frac{4}{3}\left(2^{2k-4} - 1\right)$$

implying that $|R_{2k} - R_{2k-1}| \le 2^{2k-2} - \frac{4}{3}\left(2^{2k-4} - 1\right) = \frac{2^{2k-1}+4}{3}$.

Thus, Lemma 2 is proved. $\square$

**Lemma 3.** *For $i \ge 4$, $|R_{i+1} - R_i|$s are evaluated exactly by getting rid of the redundancies:*

*More explicitly, for $j \ge 6$, not all elements in $R_j$ are distinct. In fact, for $k \ge 3$, $R_{2k}$ has $2^{2k-4} - 2$ and $R_{2k+1}$ has $2^{2k-3} - 2$ pairs of elements which are numerically equal so that*

$$|R_{2k} - R_{2k-1}| = \frac{5 \cdot 2^{2k-4} + 10}{3}, \quad |R_{2k+1} - R_{2k}| = \frac{5 \cdot 2^{2k-3} + 8}{3}$$

**Proof.** From now on, we refer to duplicates as those pairs of polynomials or elements in $R$ which have different algebraic expressions, but becaue of our choice of $r$, they are numerically equal. In order to exactly evaluate $|R_{i+1} - R_i|$ for $i \ge 4$, we need to identify such pairs.

Thus, $R_6 - R_5$ has two pairs of duplicates, namely, $r_g^2 - r_g^5 + r_g^6$ & $r_g - r_g^2 + r_g^4 - r_g^5 + r_g^6$; $r_g - r_g^2 + r_g^5 - r_g^6$ and $r_g^2 - r_g^4 + r_g^5 - r_g^6$ because

$$r_g - r_g^2 + r_g^4 - r_g^5 + r_g^6 = r_g^2 - r_g^4 + r_g^4 - r_g^5 + r_g^6 = r_g^2 - r_g^5 + r_g^6$$

$$r_g - r_g^2 + r_g^5 - r_g^6 = r_g^2 - r_g^4 + r_g^5 - r_g^6$$

In general, $R_{2k} - R_{2k-1}$ has $2 + 4 + \cdots + 2^{2k-5}$ pairs of duplicates implying that $|R_{2k} - R_{2k-1}| = \frac{2^{2k-1}+4}{3} - (2^{2k-4} - 2) = \frac{5 \cdot 2^{2k-4} + 10}{3}$. Here, each pair in the union are disjoint sets.

Also, $R_{2k+1} - R_{2k}$ has $2 + 4 + \cdots + 2^{2k-5}$ pairs of duplicates implying that $|R_{2k+1} - R_{2k}| = \frac{2^{2k}+2}{3} - (2^{2k-3} - 2) = \frac{5 \cdot 2^{2k-3} + 8}{3}$. Again, each pair in the union are disjoint sets.

Thus, Lemma 3 is proved. $\square$

**Lemma 4.** *$\lambda$-measures of $R_{i+1} - R_i$ for $i \ge 3$ are calculated as:*

*First of all, $\lambda(R_3) = \lambda(r_g - r_g^2)(p_{10} + p_{01})$ and for $k \ge 2$,*

$$\lambda(R_{2k} - R_{2k-1}) = \lambda(R_{2k-1} - R_{2k-2})(p_{10} + p_{01}) + \lambda(r_g - r_g^2)p_{10}^{2k-3}p_{01}(p_{10} + p_{01})$$

*which equals*

$$\lambda(R_{2k} - R_{2k-1}) = \lambda\left(r_g - r_g^2\right)(p_{10} + p_{01})^{2k-2}\left[1 + \sum_{l=0}^{k-2} p_{10}^{2l+1} p_{01} (p_{10} + p_{01})^{-2l-1} - \sum_{l=1}^{k-3} p_{10}^{2l+1} p_{01} (p_{10} + p_{01})^{-2l-2}\right] \quad (4)$$

*where for $k = 2$, the last sum in the above equation is absent. Also, we have,*

$$\lambda(R_{2k+1} - R_{2k}) = \lambda(R_{2k} - R_{2k-1})(p_{10} + p_{01}) - \lambda\left(r_g - r_g^2\right) p_{10}^{2k-3} p_{01}(p_{10} + p_{01})$$

*which equals*

$$\lambda(R_{2k+1} - R_{2k}) = \lambda\left(r_g - r_g^2\right)(p_{10} + p_{01})^{2k-1}\left[1 + \sum_{l=0}^{k-2} p_{10}^{2l+1} p_{01}\left[(p_{10} + p_{01})^{-2l-1} - (p_{10} + p_{01})^{-2l-2}\right]\right] \quad (5)$$

*where for $k = 2$, $R_{2k} - R_{2k-1} = R_4 - R_3 = R_4$ and $R_{2k-1} - R_{2k-2} = R_3 - R_2 = R_3$.*

**Proof.** Recall that $R_3 = F_2(r_g^3) = \{r_g^2 - r_g^3, r_g - r_g^2 + r_g^3\}$. Then, using (2) and Proposition 3, we have

$$\lambda(R_3) = \lambda(r_g^2 - r_g^3) + \lambda(r_g - r_g^2 + r_g^3) = \lambda(r_g - r_g^2)(p_{10} + p_{01})$$

Next, we have $R_4 = F_3\left(r_g^4\right) = \{r_g^3 - r_g^4, r_g^2 - r_g^3 + r_g^4, r_g - r_g^3 + r_g^4, r_g - r_g^2 + r_g^3 - r_g^4\}$. We find $\lambda$-measures of these points by making use of (2) and Remark 2. Thus, we notice that

$$\lambda(1 - r_g + r_g^2 - r_g^3) = \lambda(r_g - r_g^3 + r_g^4) = \lambda(r_g - r_g^2)p_{10}p_{01}$$

Putting all these together, $\lambda(R_4 - R_3)$ equals

$$\lambda(r_g - r_g^2)(p_{10} + p_{01})^2 + \lambda(1 - r_g + r_g^2 - r_g^3)(p_{10} + p_{01}) = \lambda(r_g - r_g^2)(p_{10} + p_{01})^2 + \lambda(r_g - r_g^2)\,p_{10}p_{01}(p_{10} + p_{01})$$

In other words,

$$\lambda(R_4 - R_3) = \lambda(R_3)(p_{10} + p_{01}) + \lambda(r_g - r_g^2)\,p_{10}p_{01}(p_{10} + p_{01}) \quad (6)$$

It is to be noted that $R_4 \cap R_3 = \emptyset$, and so $R_4 - R_3 = R_4$ which implies $\lambda(R_4 - R_3) = \lambda(R_4)$.

Before proceeding further, we notice that $F_j\left(r_g^{j+1}\right) = g\left(F_{j-1}(r_g^j)\right) \cup f_1 \circ g\left(F_{j-1}(r_g^j)\right)$ for $j \geq 4$ and $R_{j+1} - R_j$ equals $g(R_j - R_{j-1}) \cup f_1 \circ g(R_j - R_{j-1})$ for $j \geq 5$.

However, at the next stage, we have already noticed that $R_5 \cap R_4 \neq \emptyset$, and so $R_5 - R_4 \neq R_5$. In fact,

$$R_5 = F_4\left(r_g^5\right)$$

and

$$R_5 - R_4 = F_4\left(r_g^5\right) - F_2\left(r_g^4 - r_g^5\right)$$

Now, notice that

$$F_4\left(r_g^5\right) = g(R_4 - R_3) \cup f_1 \circ g(R_4 - R_3)$$

So,

$$R_5 - R_4 = g(R_4 - R_3) \cup f_1 \circ g(R_4 - R_3) - F_2\left(r_g^4 - r_g^5\right)$$

This is the same as

$$\left[g(R_4 - R_3) - g\left(r_g - r_g^3 + r_g^4\right)\right] \cup \left[f_1 \circ g(R_4 - R_3) - f_1 \circ g\left(r_g - r_g^3 + r_g^4\right)\right]$$

Since $g(R_4 - R_3) - g\left(r_g - r_g^3 + r_g^4\right)$ and $f_1 \circ g(R_4 - R_3) - f_1 \circ g\left(r_g - r_g^3 + r_g^4\right)$ do not have overlaps, we deduce that

$$\lambda(R_5 - R_4) = \lambda(R_4 - R_3)(p_{10} + p_{01}) - \lambda\left(r_g - r_g^2\right) p_{10}p_{01}(p_{10} + p_{01}) \tag{7}$$

which equals

$$\lambda(R_5 - R_4) = \lambda\left(r_g - r_g^2\right)(p_{10} + p_{01})^3 + \lambda(r_g - r_g^2) \, p_{10}p_{01}(p_{10} + p_{01})^2 - \lambda(r_g - r_g^2) \, p_{10}p_{01}(p_{10} + p_{01}) \tag{8}$$

Thus, from Equations (6) and (8), we observe that Lemma 4 is proved for $k = 2$. For general $k$, one can use induction on $k$ and carefully sort out the issues with the duplicates to complete the proof of the lemma. $\square$

**Lemma 5.** *Finally, we calculate $\lambda$-measure of R:*

$$\lambda(R) = \sum_{j=3}^{\infty} \lambda(R_j - R_{j-1}) = \sum_{j=1}^{\infty} \lambda\left(r_g - r_g^2\right)(p_{10} + p_{01})^j = \lambda\left(r_g - r_g^2\right) \cdot \frac{p_{10} + p_{01}}{1 - p_{10} - p_{01}}$$

*where we put $R_2 = \varnothing$.*

**Proof.** Using (4) and (5) for $k \geq 2$, Lemma 5 follows trivially and the proof of the theorem is complete. $\square$

## 5. Concluding Remarks

In the present context, it is interesting to recall an older problem, first introduced in [7]. It is as follows: consider the very simple situation of a $\mu$ that is supported on exactly two $2 \times 2$ stochastic matrices, namely, $\begin{pmatrix} a_1 & 1 - a_1 \\ b_1 & 1 - b_1 \end{pmatrix}$ and $\begin{pmatrix} a_2 & 1 - a_2 \\ b_2 & 1 - b_2 \end{pmatrix}$ with $a_i > b_i$ for $i = 1, 2$. Let the $\mu$-masses at these two points be $p$ and $1 - p$, respectively, where $0 < p < 1$. Let $\lambda$ be the weak limit of the convolution sequence $\mu^n$. What is the nature of $\lambda$? If we denote $a_1 - b_1 = s$ and $a_2 - b_2 = t$, then, in [12], some partial solution to this problem was mentioned. In the special case scenario when $s = t$ and $p = \frac{1}{2}$, it was observed in [13] that it is precisely the case of Bernoulli convolutions. In fact, the following proposition is stated in [13]:

**Proposition 5.** *Let $\mu$ be a probability measure giving equal mass to the matrices $\begin{pmatrix} a_1 & 1 - a_1 \\ b_1 & 1 - b_1 \end{pmatrix}$ and $\begin{pmatrix} a_2 & 1 - a_2 \\ b_2 & 1 - b_2 \end{pmatrix}$ with $a_i > b_i$ for $i = 1, 2$. Let, say, $a_1 - b_1 = a_2 - b_2 = t$. Then, the limiting measure $\lambda$ of the convolution sequence $\mu^n$ is absolutely continuous (where the limt $\lambda$ is identified as a probability on $[0, 1]$) iff the law of $\sum_{n=0}^{\infty} t^n \epsilon_n$ is absolutely continuous where $\epsilon_n$'s are i.i.d. $+1$ and $-1$ with equal probabilities.*

Although the century old problem of Bernoulli convolutions was finally solved in [14], there had been a lot of previous studies at various times in different directions in spite of it being apparently a simple problem with $\mu$ concentrated on two points only. Thus, it is quite possible that under our current set up of $\mu$ being concentrated on four matrices with $\frac{1}{2} < r < 1$, the problem may be at least as challenging as the Bernoulli convolution problem.

We bring in the context of Bernoulli convolutions here to make readers aware that for a nontrivial $\frac{1}{2} < r < 1$, one needs to explore a number of ideas to proceed towards a complete solution for our problem.

**Funding:** This research received no external funding.

**Data Availability Statement:** No new data were created or analyzed in this study. Data sharing is not applicable to this article.

**Acknowledgments:** I acknowledge my affiliation, the University of Texas Rio Grande Valley, for allowing me to use the university office, university library, and university computers for conducting my research.

**Conflicts of Interest:** The author declares no conflict of interest.

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
