# Peer review of "Limit Distributions of Products of Independent and Identically Distributed Random 2 × 2 Stochastic Matrices: A Treatment with the Reciprocal of the Golden Ratio"

_mathematics, doi:10.3390/math11244993_

Round 1

Reviewer 1 Report

Comments and Suggestions for Authors

Please find my comments in the file attached.

Author Response

  • As advised by Reviewer 1, the convention stated after the statement of the main theorem has been moved. It now appears before (lines 64-69) the statement of the main theorem (lines 87-90).
  • The lines “Although, the Century old…” has been corrected as advised by Reviewer 1.
  • While responding to another reviewer, the Introduction has been elaborated a bit more by including some lines from a previous paper by the current author. Accordingly, two more references were included:
  • Chamayou, J.F. and Letac, G. (1994): A transient random walk on stochastic matrices with Dirichlet distributions, Ann. Prob., 22, 424-430.
  • Chassaing, P., Letac, G. and Mora, M. (1985): Brocot sequences and random walks in SL(2,R), Springer LNM, 1034, p.37-50.

Also, to incorporate the advice by a third reviewer, we have included one more reference to emphasize the importance of golden ratio in the modern era:

  • Jaeger, S. (2021): The Golden Ratio in Machine Learning, IEEE Applied Imagery Pattern Recognition Workshop (AIPR), Washington, DC, USA, 2021, pp. 1-7, doi: 10.1109/AIPR52630.2021.9762080.

Reviewer 2 Report

Comments and Suggestions for Authors

attached

Author Response

As advised by Reviewer 2, the Introduction has been elaborated to provide better explanation by reiterating some statements from the previous paper of the current author. Accordingly, two more references were included:

  • Chamayou, J.F. and Letac, G. (1994): A transient random walk on stochastic matrices with Dirichlet distributions, Ann. Prob., 22, 424-430.
  • Chassaing, P., Letac, G. and Mora, M. (1985): Brocot sequences and random walks in SL(2,R), Springer LNM, 1034, p.37-50.

Also, to incorporate the advice by a third reviewer, we have included one more reference to emphasize the importance of golden ratio in the modern era:

  • Jaeger, S. (2021): The Golden Ratio in Machine Learning, IEEE Applied Imagery Pattern Recognition Workshop (AIPR), Washington, DC, USA, 2021, pp. 1-7, doi: 10.1109/AIPR52630.2021.9762080.

  1. We explained that means closure of the set , not the complement.
  2. Although we state topology, it is understood to be the topology after intersecting with the space of all stochastic matrices.
  3. To make it clearer, we have now used ‘Situation 1’ and ‘Situation 2’ as examples of situations mentioned before the colon (now at the end of line 34).
  4. I think with more details added from the previous paper and with the explanation of as the closure of , lines 30-34 should make it more understandable.  
  5. As advised by another reviewer, the convention that was originally after the statement of main theorem, now appears (lines 64-69) before the statement of the main theorem (lines 87-90). This will now make the notation
  6. It was a typo. Actually, itself is the Bernoulli distribution, it does not follow Bernoulli distribution.
  7. This was also a typo. It has been corrected to ‘strictly’. (lines 87 – 90).

Reviewer 3 Report

Comments and Suggestions for Authors

Comments on the Quality of English Language

As indicated in the report, the first sentence on the last page of the manuscript (lines 308-310), is incomplete. This needs to be corrected. But for that, the presentation is good.  

Author Response

  • There were already some lines in section 5 giving some reasoning behind the author considering the case where is the reciprocal of the golden ratio. The author has now moved these lines to the Introduction. These lines may serve as an answer to the question by Reviewer 3. Also, to elaborate as to why the golden ratio is important, the author has included a very recent paper in the reference where one can find application of the golden ratio in machine learning:

  • Jaeger, S. (2021): The Golden Ratio in Machine Learning, IEEE Applied Imagery Pattern Recognition Workshop (AIPR), Washington, DC, USA, 2021, pp. 1-7, doi: 10.1109/AIPR52630.2021.9762080.

  • While responding to another reviewer, the Introduction has been elaborated a bit more by including some lines from a previous paper by the current author. Accordingly, two more references were included:
  • Chamayou, J.F. and Letac, G. (1994): A transient random walk on stochastic matrices with Dirichlet distributions, Ann. Prob., 22, 424-430.
  • Chassaing, P., Letac, G. and Mora, M. (1985): Brocot sequences and random walks in SL(2,R), Springer LNM, 1034, p.37-50.
  • In the present work, the case has been stated, but it was indeed proved in the previous paper by the author.
  • To provide some clue to the situation other than the golden ratio, the author briefly discussed Bernoulli Convolution in Section 5 which itself was a complicated case with just two stochastic matrices in the support of the probability measure. And the current problem is even more complex as there are four matrices now in the support.
  • The sentence has been completed as advised by Reviewer 3.

Round 2

Reviewer 2 Report

Comments and Suggestions for Authors

Round 3

Reviewer 2 Report

Comments and Suggestions for Authors

attached 

Author Response

To start with, the reviewer gives a nice description of the work that is understandable to new readers.

Then, the reviewer makes a comment that the author makes a ‘deliberate effect’ to overcomplicate matters. Here the author would like to disagree and would rather mention that sometimes complicated expressions were used as the author could not notice that an easier expression exists.

Next, I move to the specific comments by the author:

  1. The reviewer has rightly pointed out that there is some confusion here. Once again there was a typo which has now been corrected. Originally it was wrongly mentioned that has masses at the following two points:

Actually,  has masses at the following two points:

This has been fixed. Hope it is understandable now.

  1. The comma has been replaced by period.
  2. The author could not see if a comma or period is needed and so this sentence has not been modified.
  3. The reviewer has rightly pointed out that an easier expression for is:

The author could not simplify the original expression and so the following complicated expression was provided:

Now the manuscript has the expression provided by the reviewer.

  1. This has been taken care of.
  2. This has been taken care of.